# Indeterminate Probability Neural Network

## Abstract

We propose a new general model called **IPNN** – **I**ndeterminate **P**robability **N**eural **N**etwork, which combines neural network and probability theory together. In the classical probability theory, the calculation of probability is based on the occurrence of events, which is hardly used in current neural networks. In this paper, we propose a new general probability theory, which is an extension of classical probability theory, and makes classical probability theory a special case to our theory. With this new theory, some intractable probability problems have now become tractable (analytical solution). Besides, for our proposed neural network framework, the output of neural network is defined as probability events, and based on the statistical analysis of these events, the inference model for classification task is deduced. IPNN shows new property: It can perform unsupervised clustering while doing classification. Besides, IPNN is capable of making very large classification with very small neural network, e.g. model with 100 output nodes can classify 10 billion categories. Theoretical advantages are reflected in experimental results.

## 1 Introduction

Humans can distinguish at least 30,000 basic object categories [1], classification of all these would have two challenges: It requires huge well-labeled images; Model with softmax for large scaled datasets is computationally expensive. Zero-Shot Learning – ZSL [2, 3] method provides an idea for solving the first problem, which is an attribute-based classification method. ZSL performs object detection based on a human-specified high-level description of the target object instead of training images, like shape, color or even geographic information. But labelling of attributes still needs great efforts and expert experience. Hierarchical softmax can solve the computationally expensive problem, but the performance degrades as the number of classes increase [4].

Probability theory has not only achieved great successes in the classical area, such as Naïve Bayesian method [5], but also in deep neural networks (VAE [6], ZSL, etc.) over the last years. However, both have their shortages: Classical probability can not extract features from samples; For neural networks, the extracted features are usually abstract and cannot be directly used for numerical probability calculation. What if we combine them?

There are already some combinations of neural network and bayesian approach, such as probability distribution recognition [7, 8], Bayesian approach are used to improve the accuracy of neural modeling [9], etc. However, current combinations do not take advantages of ZSL method.

We propose an approach to solve the mentioned problems, and our contributions are as follows:

- We propose a new general probability theory – indeterminate probability theory, which is an extension of classical probability theory, and makes classical probability theory a special case to our theory. The proposed general tractable Equation (12) is analytical solutions even for some intractable probability calculation problems.

- With this new theory, CIPNN [10] has found the analytical solution for the posterior calculation of continuous latent variables, which was regarded as intractable [6, 11]. Besides, CIPNN applied our theory and proposed a general auto encoder (CIPAE), the decoder part is not a neural network and uses a fully probabilistic inference model for the first time.

- We propose a novel unified combination of (indeterminate) probability theory and deep neural network. The neural network is used to extract attributes which are defined as discrete random variables, and the inference model for classification task is derived. Besides, these attributes do not need to be labeled in advance.

The rest of this paper is organized as follows: In Section 2, related works are discussed. In Section 3, we first introduce a coin toss game as example of human cognition to explain the core idea of IPNN. In Section 4, the indeterminate probability theory and IPNN is proposed. In Section 5, the training strategy is discussed. In Section 6, we evaluate IPNN and make an impact analysis on its hyper-parameters. Finally, we conclude the paper in Section 7.

## 2 Related Work

**Tractable Probabilistic Models.** There are a large family of tractable models including probabilistic circuits [12, 13], arithmetic circuits [14, 15], sum-product networks [16], cutset networks [17], and-or search spaces [18], and probabilistic sentential decision diagrams [19]. The analytical solution of a probability calculation is defined as occurrence, $P(A = a) = \frac{\text{number of event } (A=a) \text{ occurs}}{\text{number of random experiments}}$, which is however not focused in these models. Our proposed IPNN is fully based on event occurrence and is an analytical solution.

**Deep Latent Variable Models.** DLVMs are probabilistic models and can refer to the use of neural networks to perform latent variable inference [20]. Currently, the posterior calculation of continuous latent variables is regarded as intractable [11], VAEs [6, 21–23] use variational inference method [24] as approximate solutions. Our proposed IPNN is one DLVM with discrete latent variables and the intractable posterior calculation is now analytically solved with our proposed theory.

## 3 Background

Let's first introduce a small game – coin toss: a child and an adult are observing the outcomes of each coin toss and record the results independently (heads or tails), the child can't always record the results correctly and the adult can record it correctly, in addition, the records of the child are also observed by the adult. After several coin tosses, the question now is, suppose the adult is not allowed to watch the next coin toss, what is the probability of his inference outcome of next coin toss via the child's record?

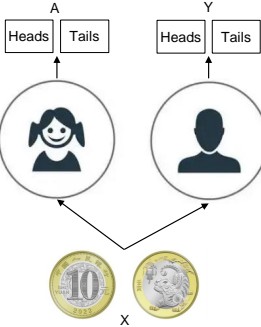

Figure 1: Example of coin toss game.

Table 1: Example of 10 times coin toss outcomes

| Experiment | Truth | A | Y |
|---|---|---|---|
| $X = x_1$ | $hd$ | $A = hd$ | $Y = hd$ |
| $X = x_2$ | $hd$ | $A = hd$ | $Y = hd$ |
| $X = x_3$ | $hd$ | $A = hd$ | $Y = hd$ |
| $X = x_4$ | $hd$ | $A = hd$ | $Y = hd$ |
| $X = x_5$ | $hd$ | $\boldsymbol{A = tl}$ | $Y = hd$ |
| $X = x_6$ | $tl$ | $A = tl$ | $Y = tl$ |
| $X = x_7$ | $tl$ | $A = tl$ | $Y = tl$ |
| $X = x_8$ | $tl$ | $A = tl$ | $Y = tl$ |
| $X = x_9$ | $tl$ | $A = tl$ | $Y = tl$ |
| $X = x_{10}$ | $tl$ | $A = tl$ | $Y = tl$ |
| $X = x_{11}$ | $hd$ | $A = ?$ | $Y = ?$ |

As shown in Figure 1, random variables X is the random experiment itself, and $X = x_k$ represent the $k^{th}$ random experiment. Y and A are defined to represent the adult's record and the child's record,

respectively. And $hd, tl$ is for heads and tails. For example, after 10 coin tosses, the records are shown in Table 1.

We formulate X compactly with the ground truth, as shown in Table 2.

Table 2: The adult's and child's records: $P(Y|X)$ and $P(A|X)$

| $\frac{\#(Y,X)}{\#(X)}$ | $Y = hd$ | $Y = tl$ | $\frac{\#(A,X)}{\#(X)}$ | $A = hd$ | $A = tl$ |
|---|---|---|---|---|---|
| $X = hd$ | 5/5 | 0 | $X = hd$ | 4/5 | 1/5 |
| $X = tl$ | 0 | 5/5 | $X = tl$ | 0 | 5/5 |

Through the adult's record Y and the child's records A, we can calculate $P(Y|A)$, as shown in Table 3. We define this process as observation phase.

For next coin toss ($X = x_{11}$), the question of this game is formulated as calculation of the probability $P^A(Y|X)$, superscript A indicates that Y is inferred via record A, not directly observed by the adult. For example, given the next coin toss $X = hd = x_{11}$, the child's record has then two situations: $P(A = hd|X = hd = x_{11}) = 4/5$ and $P(A = tl|X = hd = x_{11}) = 1/5$. With the adult's observation of the child's records, we have $P(Y = hd|A = hd) = 4/4$ and $P(Y = hd|A = tl) = 1/6$. Therefore, given next coin toss $X = hd = x_{11}$, $P^A(Y = hd|X = hd = x_{11})$ is the summation of these two situations: $\frac{4}{5} \cdot \frac{4}{4} + \frac{1}{5} \cdot \frac{1}{6}$. Table 3 answers the above mentioned question.

Table 3: Results of observation and inference phase: $P(Y|A)$ and $P^A(Y|X)$

| $\frac{\#(Y,A)}{\#(A)}$ | $Y = hd$ | $Y = tl$ | $\sum_A \left( \frac{\#(A,X)}{\#X} \cdot \frac{\#(Y,A)}{\#A} \right)$ | $Y = hd$ | $Y = tl$ |
|---|---|---|---|---|---|
| $A = hd$ | 4/4 | 0 | $X = hd = x_{11}$ | $\frac{4}{5} \cdot \frac{4}{4} + \frac{1}{5} \cdot \frac{1}{6}$ | $\frac{4}{5} \cdot 0 + \frac{1}{5} \cdot \frac{5}{6}$ |
| $A = tl$ | 1/6 | 5/6 | $X = tl = x_{11}$ | $0 \cdot \frac{4}{4} + \frac{5}{5} \cdot \frac{1}{6}$ | $0 \cdot 0 + \frac{5}{5} \cdot \frac{5}{6}$ |

Let's go one step further, we can find that even the child's record is written in unknown language (e.g. $A \in \{ZHENG, FAN\}$), Table 3 can still be calculated by the man. The same is true if the child's record is written from the perspective of attributes, such as color, shape, etc.

Hence, if we substitute the child with a neural network and regard the adult's record as the sample labels, although the representation of the model outputs is unknown, the labels of input samples can still be inferred from these outputs. This is the core idea of IPNN.

# 4 Indeterminate Probability Theory

In this section, we propose a new general probability theory, which is derived from IPNN – a neural network with discrete deep latent variables.

## 4.1 IPNN Model Architecture

Let $X \in \{x_1, x_2, \ldots, x_n\}$ be training samples ($X = x_k$ is understood as $k^{th}$ random experiment – select one train sample.) and $Y \in \{y_1, y_2, \ldots, y_m\}$ consists of $m$ discrete labels (or classes), $P(y_l|x_k) = y_l(k) \in \{0, 1\}$ describes the label of sample $x_k$. For prediction, we calculate the posterior of the label for a given new input sample $x_{n+1}$, it is formulated as $P^{\mathbb{A}}(y_l \mid x_{n+1})$, superscript $\mathbb{A}$ stands for the medium – model outputs, via which we can infer label $y_l$, $l = 1, 2, \ldots, m$. After $P^{\mathbb{A}}(y_l \mid x_{n+1})$ is calculated, the $y_l$ with maximum posterior is the predicted label.

Figure 2a shows IPNN model architecture, the output neurons of a general neural network (FFN, CNN, Resnet [25], Transformer [26], Pretrained-Models [27], etc.) is split into N unequal/equal parts, the split shape is marked as Equation (1), hence, the number of output neurons is the summation of the split shape $\sum_{j=1}^{N} M_j$. Next, each split part is passed to 'softmax', so the output neurons can be defined as discrete random variable $A^j \in \left\{ a_1^j, a_2^j, \ldots, a_{M_j}^j \right\}, j =$

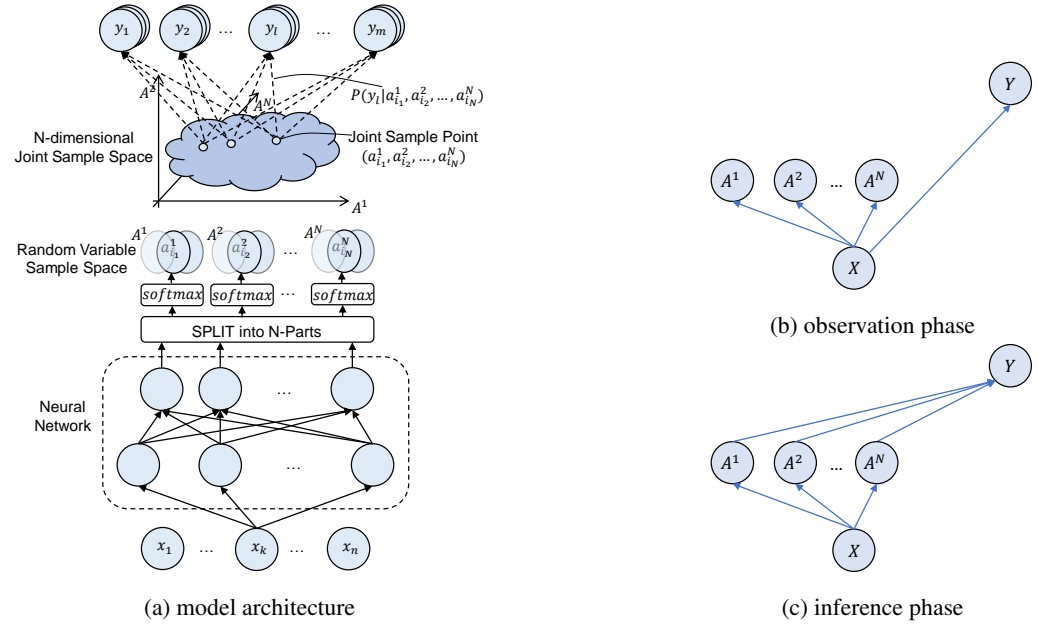

(a) model architecture

(b) observation phase

(c) inference phase

Figure 2: IPNN. (a) $P\left(y_l | a_{i_1}^1, a_{i_2}^2, \ldots, a_{i_N}^N\right)$ is statistically calculated, not model weights. (b, c) Independence illustration with Bayesian network.

$1, 2, \ldots, N$, and each neuron in $A^j$ is regarded as an event. After that, all the random variables together form the N-dimensional joint sample space, marked as $\mathbb{A} = (A^1, A^2, \ldots, A^N)$, and all the joint sample points are fully connected with all labels $Y \in \{y_1, y_2, \ldots, y_m\}$ via conditional probability $P\left(Y = y_l | A^1 = a_{i_1}^1, A^2 = a_{i_2}^2, \ldots, A^N = a_{i_N}^N\right)$, or more compactly written as $P\left(y_l | a_{i_1}^1, a_{i_2}^2, \ldots, a_{i_N}^N\right)$[1][2].

$$\text{Split shape} := \{M_1, M_2, \ldots, M_N\} \tag{1}$$

## 4.2 Definition of Indeterminate Probability

In classical probability theory, perform a random experiment (or given a sample $x_k$), the event or joint event has only two states: happened or not happened. However, for IPNN, the model only outputs the probability of an event state and its state is indeterminate, that's why this paper is called IPNN. This difference makes the calculation of probability (especially joint probability) also different. Equation (2) and Equation (3) will later formulate this difference.

Given an input sample $x_k$ (perform the $k^{th}$ random experiment), with Assumption 1 the indeterminate probability (model outputs) is defined as:

$$P\left(a_{i_j}^j \mid x_k\right) = \alpha_{i_j}^j(k) \tag{2}$$

**Assumption 1.** *Given an input sample $X = x_k$, IF $\sum_{i_j=1}^{M_j} \alpha_{i_j}^j(k) = 1$ and $\alpha_{i_j}^j(k) \in [0, 1], k = 1, 2, \ldots, n$. THEN, $\left\{a_1^j, a_2^j, \ldots, a_{M_j}^j\right\}$ can be regarded as collectively exhaustive and exclusive events set, they are partitions of the sample space of random variable $A^j, j = 1, 2, \ldots, N$.*

In classical probability, $\alpha_{i_j}^j(k) \in \{0, 1\}$, which indicates the state of event is 0 or 1.

For joint event, given $x_k$, using Assumption 2 and Equation (2), the joint indeterminate probability is formulated as:

---

[1]All the probability is formulated compactly in this paper.

[2]Reading symbols see Appendix G.

$$P\left(a_{i_1}^1, a_{i_2}^2, \ldots, a_{i_N}^N \mid x_k\right) = \prod_{j=1}^N \alpha_{i_j}^j(k) \tag{3}$$

**Assumption 2.** *Given an input sample $X = x_k$, $A^1, A^2, \ldots, A^N$ is mutually independent.*

Where it can be easily proved,

$$\sum_{\mathbb{A}}\left(\prod_{j=1}^N \alpha_{i_j}^j(k)\right) = 1, k = 1, 2, \ldots, n. \tag{4}$$

In classical probability, $\prod_{j=1}^N \alpha_{i_j}^j(k) \in \{0, 1\}$, which indicates the state of joint event is 0 or 1.

Equation (2) and Equation (3) describes the uncertainty of the state of event $\left(A^j = a_{i_j}^j\right)$ and joint event $\left(A^1 = a_{i_1}^1, A^2 = a_{i_2}^2, \ldots, A^N = a_{i_N}^N\right)$.

## 4.3 Observation Phase

In observation phase, the relationship between all random variables $A^1, A^2, \ldots, A^N$ and $Y$ is established after the whole observations, it is formulated as:

$$P\left(y_l \mid a_{i_1}^1, a_{i_2}^2, \ldots, a_{i_N}^N\right) = \frac{P\left(y_l, a_{i_1}^1, a_{i_2}^2, \ldots, a_{i_N}^N\right)}{P\left(a_{i_1}^1, a_{i_2}^2, \ldots, a_{i_N}^N\right)} \tag{5}$$

Because the state of joint event is not determinate in IPNN, we cannot count its occurrence like classical probability. Hence, the joint probability is calculated according to total probability theorem over all samples $X = (x_1, x_2, \ldots, x_n)$, and with Equation (3) we have:

$$P\left(a_{i_1}^1, a_{i_2}^2, \ldots, a_{i_N}^N\right) = \sum_{k=1}^n \left(P\left(a_{i_1}^1, a_{i_2}^2, \ldots, a_{i_N}^N \mid x_k\right) \cdot P(x_k)\right)$$
$$= \sum_{k=1}^n \left(\prod_{j=1}^N P\left(a_{i_j}^j \mid x_k\right) \cdot P(x_k)\right) = \frac{\sum_{k=1}^n \left(\prod_{j=1}^N \alpha_{i_j}^j(k)\right)}{n} \tag{6}$$

Because $Y = y_l$ is sample label and $A^j = a_{i_j}^j$ comes from model, it means $A^j$ and Y come from different observer, so we can have Assumption 3 (see Figure 2c).

**Assumption 3.** *Given an input sample $X = x_k$, $A^j$ and Y is mutually independent in observation phase, $j = 1, 2, \ldots, N$.*

Therefore, according to total probability theorem, Equation (3) and the above assumption, we derive:

$$P\left(y_l, a_{i_1}^1, a_{i_2}^2, \ldots, a_{i_N}^N\right) = \sum_{k=1}^n \left(P\left(y_l, a_{i_1}^1, a_{i_2}^2, \ldots, a_{i_N}^N \mid x_k\right) \cdot P(x_k)\right)$$
$$= \sum_{k=1}^n \left(P\left(y_l \mid x_k\right) \cdot \prod_{j=1}^N P\left(a_{i_j}^j \mid x_k\right) \cdot P(x_k)\right) \tag{7}$$
$$= \frac{\sum_{k=1}^n \left(y_l(k) \cdot \prod_{j=1}^N \alpha_{i_j}^j(k)\right)}{n}$$

Substitute Equation (6) and Equation (7) into Equation (5), we have:

$$P\left(y_l | a_{i_1}^1, a_{i_2}^2, \ldots, a_{i_N}^N\right) = \frac{\sum_{k=1}^n \left(y_l(k) \cdot \prod_{j=1}^N \alpha_{i_j}^j(k)\right)}{\sum_{k=1}^n \left(\prod_{j=1}^N \alpha_{i_j}^j(k)\right)} \tag{8}$$

Where it can be proved,

$$\sum_{l=1}^m P\left(y_l \mid a_{i_1}^1, a_{i_2}^2, \ldots, a_{i_N}^N\right) = 1 \tag{9}$$

### 4.4 Inference Phase

Given $A^j$, with Equation (8) (passed experience) label $y_l$ can be inferred, this inferred $y_l$ has no pointing to any specific sample $x_k$, incl. also new input sample $x_{n+1}$, see Figure 2b. So we can have following assumption:

**Assumption 4.** *Given $A^j$, $X$ and $Y$ is mutually independent in inference phase, $j = 1, 2, \ldots, N$.*

Therefore, given a new input sample $X = x_{n+1}$, according to total probability theorem over joint sample space $\left(a_{i_1}^1, a_{i_2}^2, \ldots, a_{i_N}^N\right) \in \mathbb{A}$, with Assumption 4, Equation (3) and Equation (8), we have:

$$
\begin{aligned}
P^{\mathbb{A}}\left(y_l \mid x_{n+1}\right) &= \sum_{\mathbb{A}} \left( P\left(y_l, a_{i_1}^1, a_{i_2}^2, \ldots, a_{i_N}^N \mid x_{n+1}\right)\right) \\
&= \sum_{\mathbb{A}} \left( P\left(y_l \mid a_{i_1}^1, a_{i_2}^2, \ldots, a_{i_N}^N\right) \cdot P\left(a_{i_1}^1, a_{i_2}^2, \ldots, a_{i_N}^N \mid x_{n+1}\right)\right) \\
&= \sum_{\mathbb{A}} \left( \frac{\sum_{k=1}^n \left(y_l(k) \cdot \prod_{j=1}^N \alpha_{i_j}^j(k)\right)}{\sum_{k=1}^n \left(\prod_{j=1}^N \alpha_{i_j}^j(k)\right)} \cdot \prod_{j=1}^N \alpha_{i_j}^j(n+1)\right)
\end{aligned}
\tag{10}
$$

And the maximum posterior is the predicted label of an input sample:

$$
\hat{y} := \underset{l \in \{1,2,\ldots,m\}}{\arg\max} \ P^{\mathbb{A}}\left(y_l \mid x_{n+1}\right)
\tag{11}
$$

### 4.5 Summary

Our most important contribution is that we propose a new general **tractable** probability Equation (10), rewritten as:

$$
\boldsymbol{P^{\mathbb{A}}\left(Y = y_l \mid X = x_{n+1}\right) =}
$$

$$
\sum_{\mathbb{A}} \left( \underbrace{\frac{\sum_{k=1}^n \left(\boldsymbol{P\left(Y = y_l \mid X = x_k\right)} \cdot \prod_{j=1}^N P\left(A^j = a_{i_j}^j \mid X = x_k\right)\right)}{\underbrace{\sum_{k=1}^n \left(\prod_{j=1}^N P\left(A^j = a_{i_j}^j \mid X = x_k\right)\right)}_{\text{Observation phase}}} \cdot \prod_{j=1}^N P\left(A^j = a_{i_j}^j \mid X = x_{n+1}\right)}_{\text{Inference phase}} \right)
\tag{12}
$$

Where X is random variable and $X = x_k$ denote the $k^{th}$ random experiment (or model input sample $x_k$), $Y$ and $A^{1:N}$ are different discrete or continuous [10] random variables. This equation can be applied to any random experiment, as long as the outcomes of random experiments are detected by some observers (neural networks, humans, or others).

Our proposed theory is derived from three our proposed conditional mutual independency assumptions, see Assumption 2 Assumption 3 and Assumption 4. However, in our opinion, these assumptions can neither be proved nor falsified, and we do not find any exceptions until now. Since this theory can not be mathematically proved, we can only validate it through experiment.

Finally, our proposed indeterminate probability theory is an extension of classical probability theory, and classical probability theory is one special case to our theory. More details to understand our theory intuitively, see Appendix B.

## 5 Training

### 5.1 Training Strategy

Given an input sample $x_t$ from a mini batch, with a minor modification of Equation (10):

$$P^{\mathbb{A}}(y_l \mid x_t) \approx \sum_{\mathbb{A}} \left( \frac{\max(H + h(\bar{t}), \epsilon)}{\max(G + g(\bar{t}), \epsilon)} \cdot \prod_{j=1}^{N} \alpha_{i_j}^{j}(t) \right) \tag{13}$$

$$h(\bar{t}) = \sum_{k=b \cdot (\bar{t}-1)+1}^{b \cdot \bar{t}} \left( y_l(k) \cdot \prod_{j=1}^{N} \alpha_{i_j}^{j}(k) \right) \tag{14}$$

$$g(\bar{t}) = \sum_{k=b \cdot (\bar{t}-1)+1}^{b \cdot \bar{t}} \left( \prod_{j=1}^{N} \alpha_{i_j}^{j}(k) \right) \tag{15}$$

$$H = \sum_{k=\max(1, \bar{t}-T)}^{\bar{t}-1} h(k), \text{for } \bar{t} = 2, 3, \ldots \tag{16}$$

$$G = \sum_{k=\max(1, \bar{t}-T)}^{\bar{t}-1} g(k), \text{for } \bar{t} = 2, 3, \ldots \tag{17}$$

Where $b$ is for batch size, $\bar{t} = \lceil \frac{t}{b} \rceil$, $t = 1, 2, \ldots, n$. Hyperparameter T is for forgetting use, i.e., $H$ and $G$ are calculated from the recent T batches. Hyper-parameter T is introduced because at beginning of training phase the calculated result with Equation (8) is not good yet. And the $\epsilon$ on the denominator is to avoid dividing zero, the $\epsilon$ on the numerator is to have an initial value of 1. Besides, $H$ and $G$ are not needed for gradient updating during back-propagation. The detailed algorithm implementation is shown in Algorithm 1.

We use cross entropy as loss function:

$$\mathcal{L} = -\sum_{l=1}^{m} \left( y_l(k) \cdot \log P^{\mathbb{A}}(y_l \mid x_t) \right) \tag{18}$$

---

**Algorithm 1** IPNN training

**Input**: A sample $x_t$ from mini-batch
**Parameter**: Split shape, forget number $T$, $\epsilon$, learning rate $\eta$.
**Output**: Posterior $P^{\mathbb{A}}(y_l \mid x_t)$

1: Declare default variables: $H, G, hList, gList$
2: **for** $\bar{t} = 1, 2, \ldots$ Until Convergence **do**
3:    Compute $h, g$ with Equation (14) and Equation (15)
4:    Record: $hList.append(h), gList.append(g)$
5:    **if** $\bar{t} > T$ **then**
6:      Forget: $H = H - hList[0], G = G - gList[0]$
7:      Remove first element from $hList, gList$
8:    **end if**
9:    Compute posterior with Equation (13): $P^{\mathbb{A}}(y_l \mid x_t)$
10:   Compute loss with Equation (18): $\mathcal{L}(\theta)$
11:   Update model parameter: $\theta = \theta - \eta \nabla \mathcal{L}(\theta)$
12:   Update for next loop: $H = H + h, G = G + g$
13: **end for**
14: **return** model and the posterior

---

With Equation (13) we can get that $P^{\mathbb{A}}(y_l \mid x_1) = 1$ for the first input sample if $y_l$ is the ground truth and batch size is 1. Therefore, for IPNN the loss may increase at the beginning and fall back again while training.

## 5.2 Multi-degree Classification (Optional)

In IPNN, the model outputs N different random variables $A^1, A^2, \ldots, A^N$, if we use part of them to form sub-joint sample spaces, we are able of doing sub classification task, the sub-joint spaces are defined as $\Lambda^1 \subset \mathbb{A}, \Lambda^2 \subset \mathbb{A}, \ldots$ The number of sub-joint sample spaces is:

$$\sum_{j=1}^{N} \binom{N}{j} = \sum_{j=1}^{N} \left( \frac{N!}{j!(N-j)!} \right) \tag{19}$$

If the input samples are additionally labeled for part of sub-joint sample spaces[3], defined as $Y^\tau \in \{y_1^\tau, y_2^\tau, \ldots, y_{m^\tau}^\tau\}$. The sub classification task can be represented as $\langle X, \Lambda^1, Y^1 \rangle, \langle X, \Lambda^2, Y^2 \rangle, \ldots$ With Equation (18) we have,

$$\mathcal{L}^\tau = -\sum_{l=1}^{m^\tau} \left( y_l^\tau(k) \cdot \log P^{\Lambda^\tau}(y_l^\tau \mid x_t) \right), \tau = 1, 2, \ldots \tag{20}$$

Together with the main loss, the overall loss is $\mathcal{L} + \mathcal{L}^1 + \mathcal{L}^2 + \ldots$ In this way, we can perform multi-degree classification task. The additional labels can guide the convergence of the joint sample spaces and speed up the training process, as discussed later in Appendix D.1.

---

[3]It is labelling of input samples, not sub-joint sample points.

### 5.3 Multi-degree Unsupervised Clustering

If there are no additional labels for the sub-joint sample spaces, the model are actually doing unsupervised clustering while training. And every sub-joint sample space describes one kind of clustering result, we have Equation (19) number of clustering situations in total.

### 5.4 Designation of Joint Sample Space

As in Appendix C proved, we have following proposition:

**Proposition 1.** *For $P(y_l|x_k) = y_l(k) \in \{0,1\}$ hard label case, IPNN converges to global minimum only when $P\left(y_l|a_{i_1}^1, a_{i_2}^2, \ldots, a_{i_N}^N\right) = 1$, for $\prod_{j=1}^N \alpha_{i_j}^j(t) > 0, i_j = 1, 2, \ldots, M_j$. In other word, each joint sample point corresponds to an unique category. However, a category can correspond to one or more joint sample points.*

**Corollary 1.** *The necessary condition of achieving the global minimum is when the split shape defined in Equation (1) satisfies: $\prod_{j=1}^N M_j \geq m$, where $m$ is the number of classes. That is, for a classification task, the number of all joint sample points is greater than the classification classes.*

Theoretically, if model with 100 output nodes are split into 10 equal parts, it can classify 10 billion categories, validation result see Appendix D.1. Besides, the unsupervised clustering (Section 5.3) depends on the input sample distributions, the split shape shall not violate from multi-degree clustering. For example, if the main attributes of one dataset shows three different colors, and your split shape is $\{2, 2, \ldots\}$, this will hinder the unsupervised clustering, in this case, the shape of one random variable is better set to 3. And as in Appendix D also analyzed, there are two local minimum situations, improper split shape will make IPNN go to local minimum.

In addition, the latter part from Proposition 1 also implies that IPNN may be able of doing further unsupervised classification task, this is beyond the scope of this discussion.

## 6 Experiments and Results

### 6.1 Unsupervised Clustering

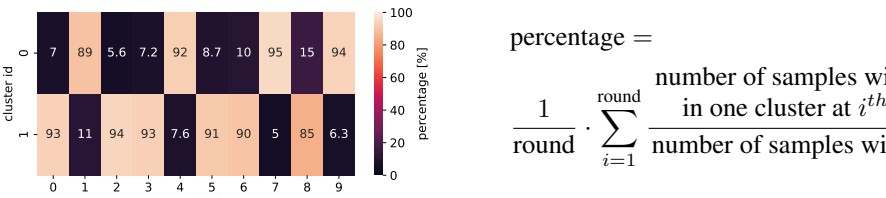

Figure 3: Unsupervised clustering results on MNIST: test accuracy $95.1 \pm 0.4$, $\epsilon = 2$, batch size $b = 64$, forget number $T = 5$, epoch is 5 per round. The test was repeated for 876 rounds with same configuration (different random seeds) in order to check the stability of clustering performance, each round clustering result is aligned using Jaccard similarity [28].

As in Section 5.3 discussed, IPNN is able of performing unsupervised clustering, we evaluate it on MNIST. The split shape is set to $\{2, 10\}$, it means we have two random variables, and the first random variable is used to divide MNIST labels $0, 1, \ldots 9$ into two clusters. The cluster results is shown in Figure 3.

We find only when $\epsilon$ in Equation (13) is set to a relative high value that IPNN prefers to put number 1,4,7,9 into one cluster and the rest into another cluster, otherwise, the clustering results is always different for each round training. The reason is unknown, our intuition is that high $\epsilon$ makes that each category catch the free joint sample point more harder, categories have similar attributes together will be more possible to catch the free joint sample point.

## 6.2 Hyper-parameter Analysis

IPNN has two import hyper-parameters: split shape and forget number T. In this section, we have analyzed it with test on MNIST, batch size is set to 64, $\epsilon = 10^{-6}$. As shown in Figure 4a, if the number of joint sample points is smaller than 10, IPNN is not able of making a full classification and its test accuracy is proportional to number of joint sample points, as number of joint sample points increases over 10, IPNN goes to global minimum for both 3 cases, this result is consistent with our analysis. However, we have exceptions, the accuracy of split shape with $\{2, 5\}$ and $\{2, 6\}$ is not high. From Figure 3 we know that for the first random variable, IPNN sometimes tends to put number 1,4,7,9 into one cluster and the rest into another cluster, so this cluster result request that the split shape need to be set minimums to $\{2, \geq 6\}$ in order to have enough free joint sample points. That's why the accuracy of split shape with $\{2, 5\}$ is not high. (For $\{2, 6\}$ case, only three numbers are in one cluster.)

Another test in Figure 4b shows that IPNN will go to local minimum as forget number T increases and cannot go to global minimum without further actions, hence, a relative small forget number T shall be found with try and error.

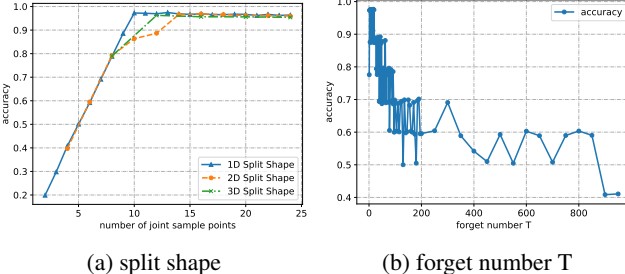

(a) split shape          (b) forget number T

Figure 4: (a) Impact Analysis of split shape with MNIST: 1D split shape is for $\{\tau\}, \tau = 2, 3, \ldots, 24$. 2D split shape is for $\{2, \tau\}, \tau = 2, 3, \ldots, 12$. 3D split shape is for $\{2, 2, \tau\}, \tau = 2, 3, \ldots, 6$. The x-axis is the number of joint sample points calculated with $\prod_{j=1}^{N} M_j$, see Equation (1).
(b) Impact Analysis of forget number T with MNIST: Split shape is $\{10\}$.

## 6.3 Evaluation on Datasets

Further results on MNIST [29], Fashion-MNIST [30], CIFAR10 [31] and STL10 [32] show that our proposed indeterminate probability theory is valid, the backbone between IPNN and 'Simple-Softmax' is the same, the last layer of the latter one is connected to softmax function. Although IPNN does not reach any SOTA, the results are very important evidences to our proposed mutual independence assumptions, see Assumption 2 Assumption 3 and Assumption 4.

Table 4: Test accuracy: split shape for all these datasets is set to $\{2, 2, 5\}$; backbone is FCN for MNIST and Fashion-MNIST, Resnet50 [25] for CIFAR10 and STL10.

| Dataset | IPNN | Simple-Softmax |
|---|---|---|
| MNIST | $95.8 \pm 0.5$ | $97.6 \pm 0.2$ |
| Fashion-MNIST | $84.5 \pm 1.0$ | $87.8 \pm 0.2$ |
| CIFAR10 | $83.6 \pm 0.5$ | $85.7 \pm 0.9$ |
| STL10 | $91.6 \pm 4.0$ | $94.7 \pm 0.7$ |

## 7 Conclusion

For a classification task, we proposed an approach to extract the attributes of input samples as random variables, and these variables are used to form a large joint sample space. After IPNN converges to global minimum, each joint sample point will correspond to an unique category, as discussed in Proposition 1. As the joint sample space increases exponentially, the classification capability of IPNN will increase accordingly.

We can then use the advantages of classical probability theory, for example, for very large joint sample space, we can use the Bayesian network approach or mutual independence among variables (see Appendix E) to simplify the model and improve the inference efficiency, in this way, a more complex Bayesian network could be built for more complex reasoning task.

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
