# OpenReview forum: "Indeterminate Probability Neural Network"
_NeurIPS.cc/2023/Conference — Submitted to NeurIPS 2023_

### Official Review · Reviewer_Xo5z · 2023-06-20

**Soundness:** 2 fair
**Presentation:** 3 good
**Contribution:** 2 fair
**Rating:** 3
**Confidence:** 3

**Summary:**

This paper proposes a new probability theory named indeterminate probability theory and a new model name Indeterminate Probability Neural Network (IPNN). For the new probability theory, it is an extension of the classical probability theory, with which some intractable probability problems become tractable (analytical solution). For the new model IPNN, it can perform unsupervised clustering while doing classification and make very large classifications with very small neural networks.

**Strengths:**

1. This paper is well organized. The intuition of the new concept of indeterminate probability is clearly demonstrated from a simple example. The theory of indeterminate probability theory is well formulated and all assumptions are clearly numbered.

2. To the best of my knowledge, the indeterminate probability theory is proposed for the first time.

**Weaknesses:**

1. The proposed new probability theory and the IPNN model are interesting contributions of this paper, but not the CIPNN. As listed in the second contribution in the introduction part, the authors claim CIPNN as the contribution of this paper. However, CIPNN is the main contribution of the authors' other papers. Thus, the authors should either delete this contribution and introduce CIPNN in other places (such as related work) or contain CIPNN in this paper (then the ICCV paper should be withdrawn).

2. There are many grammar mistakes in the use of articles. For example, 'as example' on line 46 should be 'as an example', and 'an unique category' on line 257 should be 'a unique category'.

3. On lines 110 to 112, the authors mention that the event in classical probability theory can only be happened or not happened, while for IPNN, we can have the probability of an event state. However, the authors do not consider (or demonstrate) the properties of the probability in IPNN. In classical probability theory, the happening of the event is an unbiased estimate of the probability, which means that we can approximate the true probability through plenty of experiments. But what about the probability in IPNN? What is the quality of the probability output by IPNN? Typically, the softmax output of a neural network can only be treated as a probability distribution (since the sum of all coordinates is 1) but does not indicate the true probability of one class.

4. On line 120, the authors mention that $\alpha_{ij}^j (k) \in \{ 0 , 1 \}$ in classical probability. But as defined in Eq. (2), $\alpha_{ij}^j (k)$ is a conditional probability. Then it can be any decimal between 0 and 1 if there is no more constraint. Can authors provide more detailed explanations for the claim on this line?

5. As mentioned in the abstract, IPNN 'is capable of making very large classification with very small neural network'. And the idea is using binary encoding ('the binary vector is the model inputs from 000000000000 to 111111111111, which are labeled from 0 to 4095') as mentioned in appendix D.1, which is a basic concept in information theory. But in practical implementations, one-hot encoding is preferred since it does not introduce a redundant distance between different labels. For example, the binary code 001 is closer to 000 than to 111. If binary encodings can successfully reduce the network size, this would be a great contribution to this paper. Can authors provide a more detailed introduction about output encodings (especially the comparison between binary encodings and one-hot encodings in history) and more discussions about the key tricks to making binary encodings work in IPNN?

6. As mentioned in the abstract, the indeterminate probability theory makes 'some intractable probability problems have now become tractable (analytical solution)'. It seems that the authors mix up the concept of tractable and analytical solutions. A tractable problem is a problem that can be solved with acceptable complexity (usually polynomial time and space complexity). Tractability has no relation with analytical solutions. An analytical solution can be intractable when there are exponential operations in the solution, and a tractable problem may have no analytical solution (there is no analytical expression of the error function but we can approximate the error function efficiently).

7. To my understanding, the contribution of this paper is IPNN as a new neural network (architecture, engineering trick, or training algorithm). The indeterminate probability theory is far from an extension of the classical probability theory. To extend the classical probability theory, the authors should at least formulate the new theory from measure-theoretic probability theory.

**Questions:**

See weaknesses.

**Limitations:**

Yes.

---

> ### Author Rebuttal · Authors · 2023-08-06
>
> Dear Reviewer #Xo5z,
>
> Thank you for taking the time to review our paper! Please find detailed responses in the following.
>
> **Q1: ...the authors claim CIPNN as the contribution of this paper.**
>
> A1: The purpose of highlighting CIPNN is to demonstrate that our indeterminate probability theory is general, CIPNN is an application of this theory. If you still think this is inappropriate, we will move it to the related work section.
>
>
> **Q2: the authors mention that the event in classical probability theory can only be happened or not happened...**
> **But what about the probability in IPNN?...**
> **On line 120, the authors mention that $\alpha_{ij}^j (k) \in { 0 , 1 }$ in classical probability. But as defined in Eq. (2), $\alpha_{ij}^j (k)$ is a conditional probability.**
>
> A2: We presented another simple example, we hope that will be helpful now. See reply to reviewer #1Vya in Q3.
>
>
> **Q3: For example, the binary code 001 is closer to 000 than to 111. If binary encodings can successfully reduce the network size, this would be a great contribution to this paper.**
>
> A3: In IPNN, the network size is reduced not due to binary encoding but rather to last model layer size reduction. We agree with you that one-hot encoding is preferred, we use binary encoding is only for easy of implementation, as you can see, this experiment is quite simple. We design this example only for validating the multi-degree classification and the argument in Abstract: with 100 nodes IPNN is able to classify 10,000,000,000 classes.
>
> **Q4: A tractable problem is a problem that can be solved with acceptable complexity (usually polynomial time and space complexity). Tractability has no relation with analytical solutions.**
>
> A4: Thank you very much for your detailed explanation. Our theory is:
> 1. We propose a general analytical solution of posterior, which did not exist before.
> 2. Our theoretical complexity is acceptable, as discussed in common rebuttal point 1.
>
>
> **Q5: To extend the classical probability theory, the authors should at least formulate the new theory from measure-theoretic probability theory.**
>
> A5: Our knowledge to measure-theoretic probability theory is limited, unfortunately, we don't know how to formulate our theory from it. If you could provide us with more details, we would greatly appreciate it.
>
>
> Best regards
>
> Authors

---

> > ### Comment · Reviewer_Xo5z · 2023-08-17
> >
> > Thanks for your detailed response. The authors' answers partially solve my questions, but the central contributions are still not convincing to me. It is not clear what is new to the classical probability theory, and why is the interminate probability theory needed. To my current understandings, this paper provides a method to calculate a certain posterior used in latent variable modeling, which can be done in the framework of traditional probability theory. Thus, I still think that this paper should be improved further and prefer to remain my review unchanged.

---

> > > ### Author Response · Authors · 2023-08-17
> > > **little Question.**
> > >
> > > Thanks for your response. As you have replied,  our IPNN mode can be done with the framework of traditional probability theory.
> > > Little question: How to use current probability theory to solve the problem mentioned in Sec. 3? How to use current probability theory to find the pattern of 'observer_3'? See reply to reviewer #1Vya in Q3.
> > > These examples are easy, and we hope you are interested in doing some small calculations. We look forward to your reply.

---

### Official Review · Reviewer_zXvC · 2023-07-05

**Soundness:** 2 fair
**Presentation:** 2 fair
**Contribution:** 2 fair
**Rating:** 4
**Confidence:** 4

**Summary:**

The main contribution is the proposal of the inference architecture which creates parallel softmax outputs (the authors call “splits”), which are combined into a joint soft-label space to make classification decisions under MAP rule; the joint label space can help with sub-classification tasks.

**Strengths:**

The architecture tries to create more Bayesian information before making final classification, which is a potential Bayesian neural network approach that can be developed in the future. On the other hand, this work may be inspiring to people who are interested in large-dimensional label representation.

**Weaknesses:**

1.	It is not clear to see in this paper what is new to the “classical theory of probability”;
2.	The novelty is limited.
3.	The assumptions may not be very reasonable:
The assumption 2 and 3 are ok at initialization, however when the weights are updated, A and Y are not generally independent;
The assumption 4 is very counterintuitive. Normally we have a joint distributions between X and Y, but assuming that X and Y are not independent mutually, then they are not given A; and if X and Y are independent (Y is random label for example), then they are independent given A.
4.	The main contribution of this paper is the splitting part of the architecture, creating parallel softmax outputs and combine them to make MAP decision. I think the paper should emphasize on the reasoning of this mechanism, for example, the ensemble of different likelihoods that contributes to the performance, or the geometric interpretation of the proposed label embedding/representation that makes sense.
5.	Lacking of the verification of the advantage of the new method over original softmax. After reading this paper I still do not know if this method creates more or reduce a little uncertainty in the classification compared to original softmax approach.


**Questions:**

1.	Can you compare your work with the literature on label embedding/representation?
2.	Can you make some theoretical explanation on the advantage of your methods?
3.	Can you give us better explanation on Assumption 4?


**Limitations:**

yes

---

> ### Author Rebuttal · Authors · 2023-08-06
>
> Dear Reviewer #zXvC,
>
> Thank you for taking the time to review our paper! Please find detailed responses in the following.
>
> **Q1: ...which is a potential Bayesian neural network approach.**
>
> A1. We respectfully disagree this point.
>
> **Q2: The assumptions may not be very reasonable:...The assumption 4 is very counterintuitive.**
> **Can you give us better explanation on Assumption 4?**
>
> A2:
> We suggest that you can read more details in Appendix B, which is a very intuitive aspect of understanding our theory. And Assumption 4 is the mathematical way to realize the inference.
>
> ## Core code of IPNN forward process.
>
> In addition, our indeterminate probability theory seems easy to cause confusion, you may still have many inquiries.
>
> We recommend that you can have a quick look at our implementation, you will see that from the model output 'logits', complicated calculations have been carried out until the Loss function.
>
> **If our theory is not correct, how can IPNN even converge?**
>
> ```
> # b                    --> batch size
> # y                    --> number of classification classes
> # [M_1, M_2, ..., M_N] --> split shape
>
> # inputs:
> #     logits: [b, M_1 + M_2 +, ..., M_N] # neural network outputs
> #     y_true: [b,y] # labels
> # outputs:
> #     probability: [b,y]
> #     loss
>
> logits = torch.split(logits, split_shape, dim = -1) # 43
> # Shape of variables: [[b, M_1], [b, M_2], ..., [b, M_N]]
> variables = [torch.softmax(_,dim = -1) for _ in logits] # 52
>
> # Joint sample space calculation
> # Shape of joint_variables: [b, M_1, M_2, ..., M_N]
> for i in range(len(variables)):
>     if i == 0 :
>         joint_variables = variables[i]
>     else:
>         r_ = EINSUM_CHAR[:joint_variables.dim()-1]
>         joint_variables = torch.einsum('b{},ba->b{}a'.format(r_,r_),joint_variables,variables[i]) # 112, see Eq. (3)
>
>
> # OBSERVATION PHASE
> r_ = EINSUM_CHAR[:joint_variables.dim()-1]
> num_y_joint_current = torch.einsum('b{},by->y{}'.format(r_,r_),joint_variables,y_true) # 120, see Eq. (14)
> num_joint_current = torch.sum(joint_variables,dim = 0) # 121, see Eq. (15)
>
> # numerator and denominator  of conditional probability P(Y|A^1,A^2,...,A^N)
> num_y_joint += num_y_joint_current # 167, see Eq. (16)
> num_joint += num_joint_current # 168, see Eq. (17)
>
> # Shape of prob_y_joint: [y, M_1, M_2, ..., M_N]
> prob_y_joint = num_y_joint / num_joint # 174, see Eq. (13)
>
>
> # INFERENCE PHASE
> # Shape of probability: [b,y]
> r_ = EINSUM_CHAR[:joint_variables.dim()-1]
> probability = torch.einsum('y{},b{}->by'.format(r_,r_),prob_y_joint,joint_variables) # 135, see Eq. (13)
>
>
> # loss function
> loss = cross_entropy_loss(probability,y_true) # 78 - 81, see Eq. (18)
>
> ```
>
> Where '# number' indicate the location of the code in src/ipnn.py in supplementary file.
>
> You may try to e.g. replace the 'softmax' function with 'sigmod', or make other changes, and you will find the model will no longer converge.
>
>
> **Q3: Can you compare your work with the literature on label embedding/representation?.**
>
> A3: Our theory is the analytical solution, the related analytical solutions are the classical general probability equation and Naïve Bayes, and we discussed it in Sec. 1, more details to see common rebuttal part. In our opinion, our theory stands independent from other approximate probability solutions, including label embedding and neuro-symbolic AI.
>
>
> **Q4: Lacking of the verification of the advantage of the new method over original softmax..**
>
> A4: see reply to reviewer #1Vya in Q8.
>
> **Q5: Can you make some theoretical explanation on the advantage of your methods?**
>
> A5: see reply to all reviewers, point 3.
>
> Best regards
>
> Authors

---

### Official Review · Reviewer_1Vya · 2023-07-08

**Soundness:** 2 fair
**Presentation:** 1 poor
**Contribution:** 2 fair
**Rating:** 3
**Confidence:** 2

**Summary:**

This paper proposes what I consider to be a type of neuro-symbolic AI model involving neural networks for multi-class classification problems; the authors refer to the model as an indeterminate probability neural network (IPNN). Frankly I did not fully understand the model, but the main intent appears to be a form of latent variable modeling for multi-class classification. An important line to summarize the paper is mentioned in Section 2: “our proposed IPNN is one DPVM (deep latent variable model) with discrete latent variables and the intractable posterior calculation is now analytically solved with our proposed theory”. I have reviewed a version of this paper previously, and it looks like the paper has not changed much; this is unfortunate, because I did not understand the paper clearly at that time, and I feel that I still do not understand it well.

**Strengths:**

The paper claims to blend neural networks and probability theory in a novel way; if this is true then it can be seen as an innovation in neural network modeling as well as neuro-symbolic AI. Another strength is that the paper is non-standard in that it tries to do something new around modeling multi-class classification problems.

**Weaknesses:**

In current form, the paper suffers from a number of weaknesses. A major weakness is that it remains unclear why a new theory of probability is needed in the first place. Note that the example in Section 3 can be studied using standard Bayesian modeling where X_i is the true coin toss result, A_i the adult’s reading and Y_i is the child’s reading, all for the i^{th} coin toss. Here X_i are i.i.d random variables, and A_i and Y_i are conditionally independent given X_i. Then the query of interest can be posed in terms of these random variables. I did not understand what the new theory is and why it is even needed. Also, I find it hard to follow the paper and feel it is not appropriately positioned in the literature. For instance, to me, it appears that the proposed approach is a form of neuro-symbolic AI, yet this is not even mentioned in the paper. I feel there is far too much lack of clarity in the paper in general.

**Questions:**

Some questions, comments and suggestion follow:

What exactly is the “indeterminate” aspect of IPNN? Why is it needed? Could you please explain in the paper with an example?

The distinction between the submission and the work under review on CIPNN is unclear from Section 1.

The related work section is far too small – I’m sure there is a lot of other relevant work in related areas.

Am I correct in assuming that multiple labels are not allowed for a specific sample? Could you please confirm? If so, how is this constraint enforced besides just choosing the label with maximum posterior P(y_l|x_{n+1})?

The authors claim that very large classification problems can be solved with small neural networks. However, from my limited understanding of the paper, I understood that the network splits up based on the number of training samples N. How is this then a small neural network? Doesn’t it depend on N? What is the size of the neural network (in terms of number of edges) in the proposed approach, and how is it different from the baselines?

What is the distinction between observation and inference? Is this just the distinction between learning a model and deploying it for a query?

In my opinion, the experimental section does not provide much evidence for the usefulness of the proposed approach.

I noticed several typos and grammatical errors in the paper. I suggest the authors review it carefully and fix these errors.

Some references seem incomplete.

I suggest using capital letter “B” for “Bayesian”.

**Limitations:**

The authors need to write more about the limitations of the work.

---

> ### Author Rebuttal · Authors · 2023-08-06
>
> Dear Reviewer #1Vya,
>
> Thank you very much for taking the time to review our paper again, we are glad to see you again here, and we can continue with more discussions. We hope we can answer your confusions this time.
>
> **Q1: X_i is the true coin toss result, A_i the adult’s reading and Y_i is the child’s reading, all for the i^{th} coin toss...**
>
> A1: If you change it in this way, it may have a few issues: 1. $A_i$ and $A_{i+1}$ are two different random variables. 2. True coin toss result is not able to be known.(after using an observer, this will be another $Y $.) 3.Our example is quite easy, you can try to finish the calculation with your idea to see if you can get reasonable calculation results.
>
> **Q2: The related work section is far too small**
>
> A2: see reply to reviewer #zXvC in Q3.
>
>
> **Q3: Could you please explain 'indeterminate' with an example?**
>
> A3: Yes.
>
> ## What is Indeterminate Probability？
>
> | | |  |   |   |   |   |   |   |   |   |
> | :--: | :--: | :--: | :--: | :--: | :--: | :--: | :--: | :--: | :--: | :--: |
> | Random Experiments ID X | $x_{1}$ | $x_{2}$ |  $x_{3}$ |  $x_{4}$ |  $x_{5}$ |  $x_{6}$ |  $x_{7}$ |  $x_{8}$ |  $x_{9}$ |  $x_{10}$ |
> | Ground Truth | $hd$ | $hd$ | $hd$ | $hd$ | $hd$ | $tl$ | $tl$ | $tl$ | $tl$ | $tl$ |
> | Observer_1's Record $A^1$ | $A^1 = hd$ | $A^1 = hd$ | $A^1 = hd$ | $A^1 = hd$ | $A^1 = hd$ | $A^1 = tl$ | $A^1 = tl$ | $A^1 = tl$ | $A^1 = tl$ | $A^1 = tl$ |
> | Observer_1's equivalent Representation | 1, 0 | 1, 0 | 1, 0 | 1, 0 | 1, 0 | 0, 1 | 0, 1 | 0, 1 | 0, 1 | 0, 1 |
> | Observer_2's Record $A^2$ | 0.8, 0.2 | 0.7, 0.3 | 0.9, 0.1 | 0.6, 0.4 | 0.8, 0.2 | 0.1, 0.9 | 0.2, 0.8 | 0.3, 0.7 | 0.1, 0.9 | 0.2, 0.8 |
> | Observer_3's Record $z$ | $\mathcal{N}(3,1)$ | $\mathcal{N}(3,1)$ | $\mathcal{N}(3,1)$ | $\mathcal{N}(3,1)$ | $\mathcal{N}(3,1)$ | $\mathcal{N}(-3,1)$ | $\mathcal{N}(-3,1)$ | $\mathcal{N}(-3,1)$ | $\mathcal{N}(-3,1)$ | $\mathcal{N}(-3,1)$ |
> | | |  |   |   |   |   |   |   |   |   |
>
>
> **Observer_1**
>
> Let's say, observer_1 is an adult and record the outcome of coin toss always correctly, so the probability of $A^1$ can be easily calculated with the general form:
>
> $P(A^1=hd)=\frac{\text{number of }(A^1=hd)\text{ occurs}}{\text{number of random experiments}} = \frac{5}{10}$
>
>
> If we represent observer_1's record in form of $P(A^1=hd|X=x_k)$, the probability is:
>
> $P(A^1=hd)=\sum_{k=1}^{10}P(A^1=hd|X=x_k)\cdot P(X=x_k) = \frac{1+1+1+1+1+0+0+0+0+0}{10} = \frac{5}{10}$
>
> **Observer_2**
>
> However, let's say, observer_2 is a model, it takes the image of each coin toss outcome as inputs, and it's outputs are decimal values. In this case, although the ground truth of e.g. $x_3$ outcome is determinate $hd$, this outcome cannot be 100% confirmed with a model (ground truth is not able to be known). We only have the indeterminate prediction as $P(A^2=hd|X=x_3)=0.9$. How should we handle with this situation?
>
> The observer_2's record probability shall be:
>
> $P(A^2=hd)=\sum_{k=1}^{10}P(A^2=hd|X=x_k)\cdot P(X=x_k)  = \frac{0.8+0.7+0.9+0.6+0.8+0.1+0.2+0.3+0.1+0.2}{10} = \frac{4.7}{10}$
>
> This calculation result is a **combination of ground truth and observation errors**.
>
> **Observer_3**
>
> Let's say, observer_3 is a strange unknown observer, it always outputs a Gaussian distribution for each coin toss with a 'to-be-discovered' pattern. How can we find this pattern?
>
> This is the main theory contribution of CIPNN: regard continuous variables as indeterminate probability and make the inference solvable. We write it here, because this example is helpful to understand indeterminate probability more deeply.
>
> $P(z)=\sum_{k=1}^{10}P(z|X=x_k)\cdot P(X=x_k)  = \frac{5\cdot\mathcal{N}(z;3,1)+5\cdot\mathcal{N}(z;-3,1)}{10}$
>
> We get a complexer $P(z)$ distribution now, it's form is still analytical. And this distribution have two bumps, how can we know the representation of each bump mathematically? We need to use the observer_1's record $A^1$.
>
> $P(A^1=hd|z) = \frac{\sum_{k=1}^{10}P(A^1=hd|X=x_k)\cdot P(z|X=x_k)}{\sum_{k=1}^{10}P(z|X=x_k)} = \frac{5\cdot\mathcal{N}(z;3,1)\cdot 1 +5\cdot\mathcal{N}(z;-3,1)\cdot 0}{5\cdot\mathcal{N}(z;3,1)+5\cdot\mathcal{N}(z;-3,1)}= \frac{\mathcal{N}(z;3,1)}{\mathcal{N}(z;3,1)+\mathcal{N}(z;-3,1)}$
>
> For next coin toss, let $P(z|X=x_{11})=\mathcal{N}(z;3,1)$, with Eq. (12) and Monte Carlo method, we have:
>
> $P^z(A^1=hd|X=x_{11})$
>
> $=\int_{z}\left(P(A^1=hd|z)\cdot P(z|X=x_{11})\right)$
>
> $= \mathbb{E}_{z\sim P(z\mid X=x11)}P(A^1=hd|z)$
>
> $ \approx \frac{1}{C}\sum_{c=1}^{C}P(A^1=hd|z_{c}) $
>
> $= \frac{1}{C}\sum_{c=1}^{C}\left(\frac{\mathcal{N}(z_{c};3,1)}{\mathcal{N}(z_{c};3,1)+\mathcal{N}(z_{c};-3,1)} \right)$
>
> $\approx 1,  z_\{c\} \sim \mathcal{N}(z;3,1)$
>
>
> In this way, we know that the bump with mean value 3 is for $A^{1}=hd$. (In addition, Fig 5.(b,e) from CIPNN shows 10 bumps and each bump is for one category.)
>
>
> **Q4: The distinction between the submission and CIPNN is unclear .**
>
> A4: (D-) IPNN is for discrete latent variables, CIPNN is for continuous. They are two valid applications of our indeterminate probability theory.
>
>
> **Q5:  multiple labels are not allowed?**
>
> A5: Multiple labels are allowed. But you need to redefine the random variable $Y$ as: $Y^1 \in \\{0,1\\},Y^2 \in \\{0,1\\},\dots$, similar to multi binary classification task. Examples see the model CIPAE from CIPNN.
>
> **Q6: the network splits up based on the number of training samples N.**
>
> A6: the network splits up based on classification classes, as we said in Abstract, with 100 nodes IPNN is able to classify 10,000,000,000 classes.
>
> **Q7: What is the distinction between observation and inference?**
>
> A7: see reply to reviewer #zXvC in Q2.
>
> **Q8: the experimental section does not provide much evidence for the usefulness of the proposed approach.**
>
> A8: The experiments are mainly to validate our theory, this is the most important thing. Although we get very interesting results in Fig. 3, we don't have any 'usefulness' results with IPNN.
>
>
> Best regards
>
> Authors

---

> > ### Comment · Reviewer_1Vya · 2023-08-17
> > **Thanks for your response**
> >
> > Thanks for your detailed response. I will take a closer look at some of your responses, such as the example on "indeterminate".

---

### Author Rebuttal · Authors · 2023-08-05

Dear Reviewers,

We sincerely thank all of you for taking time to read our paper.

Our proposed indeterminate probability theory is far more important than our proposed IPNN model, let's firstly focus on the theory part here.

## 1. Analytical form of any general complex posterior is discovered by us.

Currently, there is no mathematical analytical form for complex posterior, we find it now, see Eq. (8) and Eq. (12). Here is a comment from ChatGPT:

    The significance of finding an analytical form for a complex posterior distribution can be comparable to groundbreaking discoveries in other scientific disciplines. It can provide a solid foundation for understanding, predicting, and making informed decisions in complex systems, leading to advancements with far-reaching impacts in various fields.

Analytical form of $P\left(Y=y_{l}\mid A^{1}=a_{i_{1}}^{1}, A^{2}=a_{i_{2}}^{2},\dots, A^{N}=a_{i_{N}}^{N}\right)$ is shown in following table:

|| | |  |
|:--: | :--: | :---: | :---------: |
|| General Form | Naïve Bayes Form |Indeterminate Probability Form|
|Equation|$\frac{\text{number of }(Y=y_{l}, A^{1}=a_{i_{1}}^{1},\dots, A^{N}=a_{i_{N}}^{N})\text{ occurs}} {\text{number of }(A^{1}=a_{i_{1}}^{1},\dots, A^{N}=a_{i_{N}}^{N})\text{ occurs}}$ | $\frac{P(Y=y_{l})\cdot\prod_{j=1}^{N}P(A^{j}=a_{i_{j}}^{j}\mid Y=y_{l})}{P(A^{1}=a_{i_{1}}^{1},\dots, A^{N}=a_{i_{N}}^{N})}$|$\frac{ {\textstyle \sum_{k=1}^{n}\left ( P(Y=y_{l}\mid X=x_{k})\cdot  {\textstyle \prod_{j=1}^{N}}P(A^{j}=a_{i_{j}}^{j}\mid X=x_{k}) \right ) } } {{\textstyle \sum_{k=1}^{n}\left ( {\textstyle \prod_{j=1}^{N}}P(A^{j}=a_{i_{j}}^{j}\mid X=x_{k})  \right ) } }$|
|Assumption|No assumption | Given $Y$, $A^{1},A^{2},\dots,A^{N}$ conditionally independent | Given $X$, $A^{1},A^{2},\dots,A^{N}$ and $Y$ conditionally independent (See our Assumption 2 and 3.) |
|Shortages|1. Not applicable if $A^{j}$ is continuous. 2. Not applicable for indeterminate case. 3. Joint sample space is exponentially large.|1. Assumption is strong. 2. $P(A^{j}=a_{i_{j}}^{j}\mid Y=y_{l})$ not always tractable.| (Joint sample space is exponentially large, but can be solved with Monte Carlo method.) |
| Space Size| $m\cdot\prod_{j}^{N}M_{j}$| $m\cdot\sum_{j}^{N}M_{j}$ | $m\cdot n\cdot N\cdot C$ ( or $m\cdot\prod_{j}^{N}M_{j}$ if Monte Carlo is not used.) |
| | |  | |

Where $C$ is for Monte Carlo number, we can set it even to 1 (as explained in CIPNN and VAE.). More symbols to see Appendix G.

Note: If $P(A^{j}=a_{i_{j}}^{j}\mid X=x_{k}) \in\\{0,1\\}$ and $P(Y=y_{l}\mid X=x_{k}) \in\\{0,1\\}$, our indeterminate probability form is mathematically equivalent to above 'General Form'. (If necessary, we can use example to explain this part in second rebuttal phase.)

Eq. (12) is for the inference with posterior, see Appendix B.


## 2. Special Random Variable X.

To any general random experiment, it always has the random variable $X$, where $X=x_k$ is for $k^{th}$ random experiment. And the following probability is always true:

$P(X=x_k) \equiv\frac{1}{n}, k = 1,2,\dots,n.$

Random variable $X$ is the experiment itself, it shall not be mixed up with other random variables.

## 3. Why Indeterminate Probability Theory is Good?

|| | |  |
|:--: | :--: | :---: | :---------: |
|| Case | Naïve Bayes |Indeterminate Probability |
|Assumption| $A^{1},A^{2},\dots,A^{N}$ independent | Given $Y$, $A^{1},A^{2},\dots,A^{N}$ conditionally independent | See our Assumption 2, 3 and 4. |
| Validity |Strongest assumption|Strong assumption| No exception|
|Assumption Range | all samples |  few samples (due to $Y=y_{l}$) |  one sample (due to $X=x_{k}$) |
|| | |  |

Let's think the independent assumption in another way. Sometimes, $A^{1},A^{2},\dots,A^{N}$ independence assumption is strong.  Nevertheless, in the case of Naïve Bayes, the whole samples are partitioned into small groups due to condition on $Y=y_{l}$, the conditional independence maybe not strong anymore. This maybe the reason why Naïve Bayes is successful for many applications.

For our Assumption 2, 3 and 4, the whole samples are partitioned into a single sample due to $X=x_{k}$, our assumptions are the most weak one.
For example, even if $A^1$ is identical to $A^2$, our independent assumption still holds true. Furthermore, we have already conducted tests with thousand of latent variables in CIPNN, these assumptions have proven to remain valid. (In IPNN, you can test with a few variables due to the exponentially large space size during the training phase, but not during the prediction phase (Monte Carlo).)


From our perspective, constructing a toy dataset that contradicts our Assumptions 2, 3, and 4 seems practically impossible.  The endeavor of finding a counterexample can be somewhat philosophical in nature, if you have interest, we can discuss it later.


Finally, if no counterexamples are found in the future, our Assumption 2,3,4 can be considered as **Mathematical Axioms**. However, it would be too early to assert them at this stage.


## 4. Discussion

Since we now view the state of event in an indeterminate way,  we have opened the door to the applicability of indeterminate probability theory in various fields:

    For instance, we can interpret a point from data clusters as indeterminate probability, then we can do supervised classification task; We can interpret the outputs of multi-models as indeterminate probability, then we can do ensemble learning related task; Even in the field of physics, with our limited understanding of the 'Uncertainty Principle', we may interpret the position of particles as indeterminate probability distribution, and then do some inference tasks. Regarding the first two examples we have conducted preliminary validations, if required, we can provide the pseudocode in second rebuttal phase.

Finally, we will correct our grammar mistakes according to your suggestions, thanks a lot.

Best regards

Authors

---

### Decision · Program_Chairs · 2023-09-21

**Decision:**

Reject

**Comment:**

After reading the author rebuttal, all three reviewers continue to have concerns about the need, utility, and accuracy of the new theory of probability that is presented here.  In addition to the concerns of the three reviewers, the meta-reviewer is concerned that there is no discussion of axioms of the new theory of probability, how these relate to the well-known axioms of probability, or why a new theory is needed.  The justification for the new theory is focused on the impact on neural network research, but we remain a long way from understanding entire neural networks as probabilistic models, only understanding them currently as telling us P(Y|X) while ignoring all the other nodes in the network.  It is not clear that the new theory of probability will improve upon the current theory or is as mathematically rigorous as the current theory.  Of particular concern is that the author rebuttal professes no claim to address measure-theoretic foundations of probability theory.  While indeed there are attempts to avoid measure theory in formulating theories of probability, doing so requires an alternative formalization that provides equivalent rigor to that provided by measure theory.  In summary, too many concerns remain at present about the proposed new theory of probability.